# Use of Prescribed Drugs to Treat Chronic Diseases during Pregnancy in Outpatient Care in Switzerland between 2014 and 2018: Descriptive Analysis of Swiss Health Care Claims Data

**DOI:** 10.3390/ijerph19031456

**Published:** 2022-01-27

**Authors:** Eva Gerbier, Sereina M. Graber, Marlene Rauch, Carole A. Marxer, Christoph R. Meier, David Baud, Ursula Winterfeld, Eva Blozik, Daniel Surbek, Julia Spoendlin, Alice Panchaud

**Affiliations:** 1Service of Pharmacy, Lausanne University Hospital and University of Lausanne, 1011 Lausanne, Switzerland; alice.panchaud@chuv.ch; 2Institute of Primary Health Care (BIHAM), University of Bern, 3012 Bern, Switzerland; 3Department of Health Sciences, Helsana Insurance Group, 8001 Zurich, Switzerland; sereina.graber@helsana.ch (S.M.G.); evaelisabeth.blozik@uzh.ch (E.B.); 4Basel Pharmacoepidemiology Unit, Division of Clinical Pharmacy and Epidemiology, Department of Pharmaceutical Sciences, University of Basel, 4031 Basel, Switzerland; marlenesusanne.rauch@usb.ch (M.R.); caroleanna.marxer@usb.ch (C.A.M.); christoph.meier@usb.ch (C.R.M.); julia.spoendlin@usb.ch (J.S.); 5Hospital Pharmacy, University Hospital Basel, 4056 Basel, Switzerland; 6Materno-Fetal and Obstetrics Research Unit, Department “Woman-Mother-Child”, Lausanne University Hospital, 1011 Lausanne, Switzerland; david.baud@chuv.ch; 7Swiss Teratogen Information Service and Clinical Pharmacology Service, Lausanne University Hospital, 1011 Lausanne, Switzerland; ursula.winterfeld@chuv.ch; 8Institute of Primary Care, University Hospital of Zurich, 8091 Zurich, Switzerland; 9Department of Obstetrics and Gynaecology, Bern University Hospital, Insel Hospital, University of Bern, 3010 Bern, Switzerland; daniel.surbek@insel.ch

**Keywords:** chronic diseases, teratogenic, fetotoxic, claims database, pregnancy, drug utilisation, electronic database

## Abstract

Evidence on the use of drugs during pregnancy in Switzerland is lacking. We aimed to evaluate the utilisation of drugs to treat chronic diseases during pregnancy in Switzerland. We identified all pregnancies (excluding abortions) in Swiss Helsana claims data (2014–2018). In those, we identified all claims for drugs to treat a chronic disease, which typically affects women of childbearing age. Potentially teratogenic/fetotoxic drugs were evaluated during specific risk periods. Results were demographically weighted relative to the Swiss population. We identified claims for ≥1 drug of interest during 22% of 369,371 weighted pregnancies. Levothyroxine was most frequently claimed (6.6%). Antihypertensives were claimed during 5.3% (3.9% nifedipine in T3). Renin–Angiotensin–Aldosterone System (RAAS) inhibitors were dispensed to 0.3/10,000 pregnancies during trimester 2 (T2) or trimester 3 (T3). Insulin was claimed during 3.5% of pregnancies, most frequently in T3 (3.3%). Exposure to psychotropic drugs was 3.8% (mostly Selective serotonin reuptake inhibitors (SSRIs)) and to drugs for obstructive airway diseases 3.6%. Traditional immunosuppressants (excluding corticosteroids) were claimed during 0.5% (mainly azathioprine and hydroxychloroquine), biologic immunosuppressants (Tumour necrosis factor-alpha (TNF-alpha) inhibitors and interleukin inhibitors) during 0.2%, and drugs to treat multiple sclerosis during 0.09% of pregnancies. Antiretrovirals were claimed during 0.15% of pregnancies. Patterns of drug claims were in line with treatment recommendations, but relatively rare events of in utero exposure to teratogenic drugs may have had severe implications for those involved.

## 1. Introduction

Pregnant women are systematically excluded from clinical trials [1,2,3,4]. Thus, most drugs are used off-label during pregnancy and contain pregnancy-related warnings [5]. Nevertheless, many pregnant women require drug treatment, increasingly also to treat chronic diseases. Of 9459 mostly European pregnant women who were included in a multinational web-based survey (2011–2012) [6], 17% reported using drugs to treat at least one chronic disease, most frequently respiratory diseases (asthma, allergy), hypothyroidism, and depression. A Danish registry-based cohort study including data on 1,362,200 deliveries between 1989 and 2013 [7] reported that 15.5% of women who delivered in 2013 had at least one recorded diagnosis from a total of 23 investigated chronic diseases. The most frequently recorded diseases were respiratory diseases (1.73%), hypothyroidism (1.50%), anxiety and personality disorders (1.33%), inflammatory bowel diseases (0.67%), diabetes (0.48%), hypertension (0.43%), and rheumatoid arthritis (0.38%). The German SNIP survey study [8], reported that among 5320 pregnant women, 1 out of 5 reported having a chronic disease. Furthermore, 27% (/215) of pregnant women in the US 2004 medical expenditure panel survey reported having a chronic disease [9].

Data on the use of drugs for chronic diseases during pregnancy in Switzerland are lacking, but it can be assumed that the prevalence of chronic diseases among pregnant women is similar to that of other European countries. Furthermore, the average maternal age in Switzerland is continuously increasing [10] (from 28.9 years in 1989 to 32.2 years in 2020). In combination with improving disease detection and management of chronic diseases, the number of women requiring pharmacological treatment also during pregnancy is increasing in Switzerland.

In a previous study [11], we evaluated the utilisation of prescription drugs dispensed in outpatient care during pregnancy to treat acute conditions and symptoms of pregnancy in Switzerland. In this study, we evaluated the utilisation of drugs to treat chronic diseases during pregnancy in a cohort of pregnant women insured with Helsana mandatory health insurance. We focused on drugs (including pre-defined potentially teratogenic/fetotoxic drugs) used to treat chronic diseases, which typically occur in women of childbearing age. Antiepileptics are not discussed in this paper since they have previously been reported [12].

## 2. Methods

### 2.1. Data Source

We conducted a descriptive study using administrative claims data of the Swiss Helsana mandatory health insurance between January 2014 and December 2018. Helsana is the largest health insurance in Switzerland, covering approximately 1.1 million individuals from all 26 Swiss cantons (approximately 15% of the Swiss population). The Helsana claims database captures information on the bundled inpatient Swiss Diagnosis Related Groups (SwissDRG) codes, the outpatient medical Tariff system (TARMED), as well as outpatient midwife billing codes. Additionally, it captures a virtually complete history of all reimbursed claims for drugs dispensed in outpatient care, which are coded as Anatomical Therapeutic Chemical codes (ATC) [13]. Besides the information on medical services per se, the database contains demographic characteristics of insured persons including sex and age.

### 2.2. Pregnancy Cohort

#### 2.2.1. Identification of Pregnancies and Delivery Date

Our study population of pregnant women has been described previously [11,12]. We captured SwissDRG, TARMED or midwife billing codes to identify deliveries (live birth or stillbirth) between 2014 and 2018. Whenever 2 or more delivery codes were recorded within 30 days, they were considered as pertaining to the same delivery and the date of the first delivery code within this time range was defined as the delivery date. In case a delivery code was recorded more than 300 days following the first code, it was considered as pertaining to a subsequent pregnancy. In case a delivery code was recorded between 30 and 300 days following a preceding delivery code, the delivery date was set according to the recorded SwissDRG delivery code (<2.5% of deliveries in Switzerland take place in an outpatient setting [14]) and pregnancies for which no SwissDRG code was recorded (0.2% of women) were excluded (Figure 1, flow chart of the unweighted study population). Therefore, one woman may have contributed several pregnancies to the cohort. Deliveries resulting in multiple births were identified identically to deliveries of singletons and were counted as one pregnancy.

#### 2.2.2. Estimation of the Date of the Last Menstrual Period and Pregnancy Trimesters

The beginning of pregnancy is not recorded in Helsana claims data. We, therefore, estimated the date of the last menstrual period (LMP) using an algorithm previously validated in US claims data [15]. According to this algorithm, the LMP was estimated at 270 days before the delivery date. If a delivery code indicated preterm delivery (<37 gestational weeks, see Appendix B Table A1 for respective SwissDRG codes), LMP was set at 245 days before the delivery date. Each pregnancy trimester (T1, T2, T3) was defined as a 90-day period, whereby T3 was shortened in case of preterm delivery (Figure 2, estimation of the LMP and pregnancy periods). We further defined a pre-pregnancy baseline period, between 90 days before LMP and LMP. All women who were not continuously insured at Helsana between the beginning of the pre-pregnancy period and delivery date were excluded from the cohort (19.5% of women).

### 2.3. Demographics and Characteristics

For each pregnancy, we extracted the age at delivery, and the year and mode of delivery (caesarean section vs. vaginal delivery, see Appendix B Table A1 for respective codes), which are presented in Table 1.

#### Exposure to Drugs

We defined drug groups to treat chronic diseases which typically occur in women of childbearing age using the ATC classification. All included drug groups and the respective ATC codes are listed in Table 2.

Within each drug group, we identified all dispensed active substances. We screened for a list of selected active substances, with known/potential adverse effects in the newborn in trimester 1 or in trimester 2/3 (Table 3, Table 4 and Table 5). This list was composed using the online teratogen information platforms, Le CRAT, (Centre de Référence sur les Agents Tératogènes; French) [16] and Embryotox, (German) [17] and by screening all warnings issued by Swissmedic (Swiss authorization and supervisory authority for drugs and medical products [18]) between 2008 and 2021. We evaluated a selection of drugs used to treat chronic diseases, which typically manifest in women of childbearing age. Thus, we did not screen for all drugs on the Swiss market, as this would have been beyond the scope of this study.

### 2.4. Statistical Analyses

We quantified the prevalence of exposure to different drug substances overall, during each pregnancy trimester separately, and during pre-pregnancy. Exposure to known teratogenic or fetotoxic substances was limited to the specific pre-defined risk periods displayed in Table 3 and Table 4, while exposure to drugs with potential adverse effects of drugs during specific pre-defined risk periods is displayed in Table 5. Prevalence of exposure was defined as the proportion of pregnancies during which at least one prescription was filled for the respective active substance, divided by the total number of enrolled pregnancies during the respective time period.

In order to represent numbers for the overall Swiss population and due to potential small biases in the Helsana data set, all results were extrapolated/weighted relative to the demographic distribution of the overall Swiss population, taking into account the stratification by calendar year, canton, age, and sex. The extrapolations/weightings were based on individual weighting factors (w_i_), which were calculated as the inverse of the sampling probability (p_i_ = N_Helsana,i_/N_Switzerland,i_) of a given stratum (i): w_i_ = 1/p_i_. The strata are defined by a woman’s demographic characteristics at the time of the delivery. The corresponding sample sizes (N_Helsana,i_, N_Switzerland,i_) for the different strata come from the risk equalization statistics.

The weighted sums (weighted number of pregnancies), weighted mean, and standard deviation of age were calculated using the survey package in R [19].

All data are anonymous, and all analyses were conducted by the Helsana Department of Health Sciences using the statistical programming language R (version 3.6.1, [20]).

### 2.5. Protocol Approvals

Ethics committee approval was not required because data used for the study were anonymous.

## 3. Results

We identified a weighted population of 369,371 pregnancies from 323,632 women, with a mean age of the pregnant woman at delivery of 32.0 years (SD 5.1). In total, 33.7% of all pregnancies resulted in caesarean section. (Table 1, unweighted study population is displayed in Appendix A. This corresponds to the average proportion of caesarean sections in Switzerland [21].

**Table 1 ijerph-19-01456-t001:** Description of the weighted study population.

Year	N of Weighted Deliveries in Our Study Population	Weighted Age at Delivery in the Cohort (Weighted sd)	Age at Delivery in Switzerland (BfS)	Weighted Percentage of Caesarean Section in the Cohort (%, 95% CI)	Percentage of Caesarean Section in Switzerland (BfS)
2014	71,933	31.96 (5.04)	31.7	34.4 (33.4, 35.5)	33.7
2015	71,844	31.97 (5.15)	31.8	34.3 (33.3, 35.4)	33.3
2016	74,149	31.93 (5.11)	31.8	33.4 (32.4, 34.5)	33.2
2017	79,610	32.06 (5.14)	31.9	33.4 (32.4, 34.6)	32.3
2018	71,836	32.15 (5.00)	32.0	33.1 (32.0, 34.2)	32.1

### 3.1. Prevalence of Drug Exposure

All unweighted numbers are displayed in the Appendix A. Figure 3 described the prevalence of pregnancies per each studied drug group during pregnancy (T1–T3). Table 2 presents the overall prevalence of claims for each drug group and for all substances within each group by descending prevalence (in the absence of a clinically relevant between different drugs of a specific drug group, only the two most frequent substances are presented).

**Table 2 ijerph-19-01456-t002:** Exposure prevalence to different drug groups and active substances during pregnancy overall and within trimester of pregnancy and pre-pregnancy separately (weighted numbers).

ATC Code	Drug Substance	Pre-Pregnancy N (/10,000)	95% CI	T1 N (/10,000)	95% CI	T2 N (/10,000)	95% CI	T3 N (/10,000)	95% CI	T1–T3 N (/10,000)	95% CI
	At least one drug from any of the following drug groups	43,179 (1169.0)	1136.0–1202.0	39,978 (1008.6)	1050.4–1114.3	41,601 (1126.3)	1093.8–1158.7	57,980 (1569.7)	1531.5–1607.9	81,715 (2212.3)	2168.4–2256.1
**H03**	**Thyroid therapy**	**10,581 (286.5)**	**270.1–302.9**	**17,570 (475.7)**	**454.3–497.0**	**19,111 (517.4)**	**495.3–539.5**	**17,677 (478.6)**	**457.3–499.8**	**24,874 (673.4)**	**648.3–698.5**
H03A	Thyroid preparations	10,223 (276.8)	260.6–292.9	17,206 (465.8)	444.7–486.9	18,834 (509.9)	488–531.8	17,470 (473)	451.8–494.1	24,418 (661.1)	636.2–686
*H03AA01*	*Levothyroxine sodium*	*10,087 (273.1)*	*257.1–289.1*	*17,081 (462.4))*	*441.4–483.4*	*18,690(506)*	*484.2–527.8*	*17,365 (470.1)*	*449–491.2*	*24,228 (655.9)*	*631.2–680.7*
*H03AA02*	*Liothyronine sodium*	*0.0 (0.0)*	*0.0–0.0*	*0 (0.0)*	*0.0–0.0*	*0 (0.0)*	*0.0–0.0*	*0 (0.0)*	*0.0–0.0*	*0 (0.0)*	*0.0–0.0*
H03B	Antithyroid preparations	388 (10.5)	7.4–13.6	378 (10.2)	7.2–13.3	284 (7.7)	5–10.4	207 (5.6)	3.4–7.8	491 (13.3)	9.7–16.9
*H03BA02*	*Propylthiouracil*	*174 (4.7)*	*2.6–6.8*	*285 (7.7)*	*5–10.5*	*137 (3.7)*	*1.8–5.6*	*88 (2.4)*	*0.9–3.8*	*328 (8.9)*	*5.9–11.8*
*H03BB01*	*Carbimazole*	*255 (6.9)*	*4.3–9.5*	*174 (4.7)*	*2.7–6.7*	*164 (4.4)*	*2.4–6.5*	*126 (3.4)*	*1.7–5.1*	*311 (8.4)*	*5.6–11.2*
**C02, C03, C07, C08, C09**	**Anti-hypertensive drugs**	**2612 (70.7)**	**62.2–79.2**	**2290 (62)**	**53.9–70.1**	**4716 (127.7)**	**116.3–139**	**17,177 (465)**	**443.5–486.6**	**19,594 (530.5)**	**507.6–553.3**
C08	Calcium channel blockers	279 (7.6)	4.7–10.4	405 (11)	7.9–14.1	2917 (79)	70.2–87.7	14,585 (394.9)	375.2–414.5	15,778 (427.2)	*406.8–447.5*
*C08CA05*	*Nifedipine*	*157 (4.3)*	2.1–6.4	*289 (7.8)*	5.2–10.5	*2827 (76.5)*	67.9–85.1	*14,423 (390.5)*	371–410	*15,531 (420.5)*	*400.3–440.7*
*C08CA01*	*Amlodipine*	*94 (2.5)*	0.8–4.3	*106 (2.9)*	1.2–4.5	*85 (2.3)*	0.7–3.9	*154 (4.2)*	2–6.3	*233 (6.3)*	*3.7–8.9*
C07	Beta-blocking agents	1651 (44.7)	37.9–51.5	1847 (50)	42.6–57.4	1860 (50.4)	43–57.8	2878 (77.9)	68.6–87.3	4193 (113.5)	*102.4–124.6*
*C07AG01*	*Labetalol*	*225 (6.1)*	3.7–8.5	*670 (18.1)*	13.3–23	*987 (26.7)*	20.8–32.6	*2019 (54.7)*	46.5–62.8	*2484 (67.2)*	*58.2–76.3*
*C07AB02*	*Metoprolol*	*555 (15)*	11.2–18.9	*640 (17.3)*	13.3–21.3	*520 (14.1)*	10.7–17.5	*521 (14.1)*	10.6–17.6	*986 (26.7)*	*21.8–31.6*
C09	RAA acting agents	610 (16.5)	12.3–20.7	227 (6.1)	3.7–8.6	105 (2.8)	1.1–4.5	72 (1.9)	0.6–3.2	341 (9.2)	*6.3–12.2*
*C09CA06*	*Candesartan*	*59 (1.6)*	0.4–2.8	*54 (1.5)*	0.3–2.6	*0 (0.0)*	0.0–0.0	*0 (0.0)*	0.0–0.0	*54 (1.5)*	*0.3–2.6*
*C09AA03*	*Lisinopril*	*87 (2.4)*	0.9–3.8	*46 (1.2)*	0.0–2.5	*30 (0.8)*	0.0–1.8	*13 (0.4)*	0.0–0.9	*58 (1.6)*	*0.2–2.9*
C03	Diuretics	*277 (7.5)*	4.7–10.3	*105 (2.8)*	1.1–4.6	*59 (1.6)*	0.4–2.8	*126 (3.4)*	1.7–5.1	*271 (7.3)*	*4.6–10*
*C03CA01*	*Furosemide*	77 (2.1)	0.7–3.5	18 (0.5)	0.0–1.0	27 (0.7)	0.0–1.6	95 (2.6)	1.0–4.1	139 (3.8)	*1.9–5.6*
*C03CA04*	*Torasemide*	*94 (2.5)*	0.6–4.4	*58 (1.6)*	0.1–3.1	*32 (0.9)*	0.0–1.8	*31 (0.8)*	0.1–1.6	*115 (3.1)*	*1.2–5.0*
C02	Antihypertensives	*58 (1.6))*	0.3–2.9	*17 (0.5)*	0.0–1.4	*17 (0.5)*	0.0–1.4	*24 (0.6)*	0.0–1.6	*24 (0.6)*	*0.0–1.6*
*C02AB01*	*Methyldopa*	*0 (0.0)*	*0.0–0.0*	*17 (0.5)*	*0.0–1.4*	*17 (0.5)*	*0.0–1.4*	*25 (0.6)*	*0.0–1.6*	*24 (0.6)*	*0.0–1.6*
*C02CA04*	*Doxazosin*	*0 (0.0)*	*0.0–0.0*	*17 (0.5)*	*0.0–1.4*	*0 (0.0)*	*0.0–0.0*	*0 (0.0)*	*0.0–0.0*	*17 (0.5)*	*0.0–1.4*
**N06A, N05A, N05AN, N05BA, N05CD, N05CF**	**Psychotropic drugs**	**15,794 (427.6)**	**407.2–448**	**9222 (249.7)**	**234.1–265.2**	**6468 (175.1)**	**162.2–188**	**6236 (168.8)**	**156.3–181.4**	**14,084 (381.3)**	**362.2–400.4**
**N06A**	**Antidepressants**	**9442 (255.6)**	239.9–271.3	**6488 (175.6)**	162.7–188.6	**4572 (123.8)**	112.9–134.6	**3940 (106.7)**	96.9–116.4	**9150 (247.7)**	**232.4–263**
N06AB	Selective serotonin reuptake inhibitors	5632 (152.5)	140.4–164.6	4143 (112.2)	101.7–122.6	3263 (88.3)	79.1–97.6	2831 (76.6)	68.3–85	5952 (161.1)	148.6–173.7
*N06AB10*	*Escitalopram*	*2313 (62.6)*	54.9–70.3	*1422 (38.5)*	32.4–44.6	*972 (26.3)*	21.5–31.1	*912 (24.7)*	20–29.4	*2054 (55.6)*	48.4–62.8
*N06AB06*	*Sertraline*	*1324 (35.8)*	30.0–41.7	*1093 (29.6)*	24–35.1	*1148 (31.1)*	25.5–36.7	*1075 (29.1)*	23.9–34.3	*1817 (49.2)*	42.2–56.2
N06AX	Other antidepressants	3871 (104.8)	94.8–114.8	2353 (63.7)	56.1–71.3	1344 (36.4)	30.7–42	1092 (29.6)	24.5–34.6	3360 (91)	81.9–100.1
*N06AX25*	*Hyperici herba*	*935 (104.8)*	20.2–30.4	*609 (16.5)*	12.6–20.4	*519 (14.1)*	10.5–17.6	*487 (13.2)*	9.7–16.6	*1291 (35)*	29.3–40.6
*N06AX16, N06AX21*	*Serotonin and noradrenaline reuptake inhibitors (duloxetine. venlafaxine)*	*1155 (31.3)*	26–36.6	*835 (22.6)*	18.2–27	*433 (11.7)*	8.6–14.9	*329 (8.9)*	6.1–11.7	*946 (25.6)*	20.8–30.4
*N06AX11,* *N06AX03*	*Alpha antagonists (mirtazapine. mianserine)*	*584 (15.8)*	*12.0–14.3*	*370 (10.0)*	*7.0–13.0*	*203 (5.5)*	*3.2–7.8*	*122 (3.3)*	*1.7–4.9*	*508 (13.8)*	10.2–17.3
N06AA	Non-selective monoamine reuptake inhibitors	574 (15.5)	11.6–19.4	396 (10.7)	7.4–14.1	138 (3.7)	1.8–5.6	137 (3.7)	1.9–5.5	506 (13.7)	10–17.4
*N06AA06*	*Trimipramine*	*321 (8.7)*	5.7–11.7	*203 (5.5)*	3.1–7.9	*53 (1.4)*	0.1–2.8	*45 (1.2)*	0.0–2.4	*238 (6.4)*	3.8–9.0
*N06AA09*	*Amitriptyline*	*163 (4.4)*	2.3–6.5	*113 (3.1)*	1.4–4.7	*78 (2.1)*	0.9–3.4	*92 (2.5)*	1.1–3.9	*181 (4.9)*	2.8–7.0
**N05A**	**Antipsychotics**	**1881 (50.9)**	**43.9–58.0**	**1099 (29.8)**	**24.3–35.2**	**699 (18.9)**	**14.8–23.1**	**761 (20.6)**	**16.2–24.9**	**1424 (38.6)**	**32.4–44.8**
*N05AH04*	*Quetiapine*	*749 (20.3)*	15.9–24.6	*506 (13.7)*	10.2–17.2	*433 (11.7)*	8.5–8.5	*456 (12.3)*	9.0–15.6	*650 (17.6)*	13.6–21.6
*N05AX12*	*Aripiprazole*	*267 (7.2))*	4.5–10	*173 (4.7)*	2.5–6.9	*108 (2.9)*	1.3–4.6	*125 (3.4)*	1.6–5.2	*251 (6.8)*	4.1–9.5
*N05AH03*	*Olanzapine*	*177 (4.8)*	2.5–7.1	*139 (3.8)*	1.5–6.0	*49 (1.3)*	0.1–2.5	*87 (2.4)*	0.9–3.8	*190 (5.1)*	2.6–7.7
*N05AX08*	*Risperidone*	*103 (2.8)*	1.2–4.4	*47 (1.3)*	0.3–2.2	*21 (0.6)*	0.0–1.2	*18 (0.5)*	0.0–1.2	*79 (2.1)*	0.8–3.4
*N05AD01*	*Haloperidol*	*70 (1.9)*	0.6–3.2	*62 (1.7)*	0.1–3.3	*28 (0.8)*	0.0–1.6	*23 (0.6)*	0.01.5	*64 (1.7)*	0.9–4.7
*N05AD08*	*Droperidol*	*390 (10.6)*	7.3–13.8	*35 (0.9)*	0.1–1.8	*12 (0.3)*	0.0–0.8	*6 (0.2)*	0.0–0.5	*53 (1.4)*	0.4–2.5
*N05AF03*	*Chlorprothixene*	*81 (2.2)*	0.8–3.6	*64 (1.7)*	0.4–3.1	*29 (0.8)*	0.0–1.7	*23 (0.6)*	0.0–1.5	*104 (2.8)*	0.4–3.1
*N05AN01*	*Lithium*	*17 (0.5)*	0.0–1.1	*20 (0.5)*	0.0–1.1	*10 (0.3)*	0.0–0.8	*10 (0.3)*	0.0–0.8	*29 (0.8)*	0.0–1.6
**N05BA, N05CD, N05CF**	**Benzodiazepine derivatives/related drugs**	**7509 (203.3)**	**188.9–217.6**	** *3492 (94.5)* **	**84.7–104.4**	** *1926 (52.1)* **	**45–59.2**	** *2180 (59)* **	**51.3–66.8**	** *6141 (166.3)* **	**153.3–179.2**
N05BA, N05CD	Benzodiazepine derivatives	6260 (169.5)	156.3–182.7	2656 (71.9)	63.4–80.4	1642 (44.5)	37.9–51	1749 (47.4)	40.4–54.3	4958 (134.2)	122.6–145.8
*N05BA06*	*Lorazepam*	*2788 (75.5)*	66.7–84.3	*1390 (37.6)*	31.4–43.9	*853 (23.1)*	18.3–27.9	*1034 (28)*	22.6–33.4	*2813 (76.2)*	67.3–85.0
*N05CD08*	*Midazolam*	*1963 (53.1)*	45.6–60.7	*360 (9.7)*	6.8–12.7	*68 (1.8)*	0.8–3.0	*113 (3.1)*	1.2–4.9	*514 (13.9)*	10.3–17.6
N05CF	Benzodiazepine related drugs	1606 (43.5)	37–50	989 (26.8)	21.4–32.1	337 (9.1)	6.1–12.0	517 (14)	10.3–17.7	1486 (40.2)	33.7–46.7
*N05CF02*	*Zolpidem*	*1425 (38.6)*	32.5–44.7	*908 (24.6))*	19.5–29.6	*333 (9)*	6.1–12.0	*511 (13.8)*	10.1–17.5	*1486 (40.2)*	31.6–44
*N05CF01*	*Zopiclone*	*200 (5.4)*	2.9–7.9	*87 (2.4)*	0.5–4.2	*4 (0.1)*	0.0–0.3	*6 (0.2)*	0.0–0.5	*97 (2.6)*	0.7–4.5
**A10**	**Anti-diabetics**	**3592 (97.2)**	**87.6–106.9**	**3227 (87.4)**	**78–96.8**	**4540 (122.9)**	**112–133.9**	**12,659 (342.7)**	**324.8–360.7**	**14,398 (389.8)**	**370.7–408.9**
A10A	Insulins and analogues	1050 (28.4)	23.2–33.7	1598 (43.3)	36.7–49.8	4060 (109.9)	99.7–120.2	12,258 (331.9)	314.3–349.5	12,752 (345.2)	327.3–363.2
A10AE	Long acting	498 (13.5)	9.8–17.2	866 (23.4)	18.5–28.4	2434 (65.9)	58.2–73.6	7328 (198.4)	185.1–211.7	7903 (214)	200.1227.8
A10AB	Fast acting	922 (25)	20.1–29.8	1281 (34.7)	28.9–28.9	2104 (57)	49.5–40.4	5812 (157.3)	145.1–169.6	6085 (164.7)	152.2–177.3
A10AC	Intermediate acting	75 (2)	0.6–3.5	203 (5.5)	3.2–7.8	1027 (27.8)	22.3–33.3	3688 (99.8)	89.9–109.8	3978 (107.7)	97.3–118.1
A10B	Blood glucose-lowering drugs	2650 (71.7)	63.5–80	1802 (48.8)	41.7–55.9	595 (16.1)	12–20.3	608 (16.5)	12.1–20.8	2243 (60.7)	52.8–68.6
*A10BA02*	*Metformin*	*2483 (67.2)*	59.2–75.2	*1726 (46.7)*	39.8–53.7	*569 (15.4)*	11.3–19.5	*541 (14.6)*	10.5–18.8	*2086 (56.5)*	48.9–64.1
*A10BJ02*	*Liraglutide*	*63 (1.7)*	0.5–2.9	*32 (0.9)*	0.1–1.6	*6 (0.2)*	0.0–0.5	*18 (0.5)*	0.0–1.2	*56 (1.5)*	0.4–2.6
*A10BB01*	*Glibenclamide*	*0 (0.0)*	0.0–0.0	*0 (0.0)*	0.0–0.0	*10 (0.3)*	0.0–0.8	*36 ()*	0.0–1.9	*46 (1.2)*	0.2–2.4
**R03**	**Obstructive airway diseases**	**7332 (198.5)**	**184.6–212.3**	**5775 (156.3)**	**143.9–168.7**	**6345 (171.8)**	**158.9–184.7**	**6077 (164.5)**	**151.9–177.2**	**13,414 (363.2)**	**344.4381.9**
R03AK	Adrenergics with glucocorticoids or other drugs (excl anticholinergics)	5115 (138.5)	126.9–150.1	3566 (96.5)	86.7–106.4	3950 (106.9)	96.7–117.1	3572 (96.7)	86.8–106.6	7925 (214.6)	200.1–229
*R03AK07*	*Formoterol and budesonide*	*3396 (91.9)*	82.7–101.2	*2219 (60.1)*	52.5–67.6	*2618 (70.9)*	62.8–78.9	*2242 (60.7)*	53.1–68.3	*5337 (144.5)*	132.8–156.1
*R03AK06*	*Salmeterol and fluticasone*	*1496 (40.5)*	33.9–47.1	*1158 (31.4)*	25.5–37.2	*1249 (33.8)*	27.7–39.9	*1199 (32.5)*	26.4–38.5	*2429 (65.8)*	57.3–74.2
R03AC	Selective B2 adrenoreceptor agonists	2714 (73.5)	64.9–82.1	2499 (67.7)	59.4–75.9	2636 (71.4)	63.1–79.6	2271 (61.5)	53.8–69.2	5947 (161)	148.3–173.7
*R03AC02*	*Salbutamol*	*2362 (63.9)*	55.8–72.1	*2132 (57.7)*	50.1–65.4	*2297 (62.2)*	54.4–70	*2046 (55.4)*	48–62.8	*5289 (143.2)*	131.2–155.2
*R03AC13*	*Formoterol*	*182 (4.9)*	3–6.9	*244 (6.6)*	4.1–9.2	*139 (3.8)*	2.1–5.5	*123 (3.3)*	1.7–5.0	*348 (9.4)*	6.4–12.5
R03BA	Inhaled Glucocorticoids	596 (16.1)	12.3–19.9	758 (20.5)	16.1–24.9	899 (24.3)	19.7–29.0	758 (20.5)	16.2–24.8	2011 (54.4)	47.3–61.6
*R03BA02*	*Budesonide*	*227 (6.1)*	3.8–8.4	*410 (11.1)*	7.8–14.4	*479 (13)*	9.7–16.3	*410 (11.1)*	8.0–14.2	*1156 (31.3)*	26.0–36.6
*R03BA05*	*Fluticasone*	*210 (5.7)*	3.3–8.1	*191(5.2)*	2.9–7.4	*269 (7.3)*	4.7–9.8	*189 (5.1)*	3.1–7.1	*514 (13.9)*	10.3–17.6
R03CC	Selective beta-2 adrenoreceptor agonists	20 (0.5)	0.0–1.2	12 (0.3)	0.0–0.8	6 (0.2)	0.0–0.5	412 (11.2)	8.2–14.1	424 (11.5)	8.4–14.5
*R03CC05*	*Hexoprenaline*	*0 (0.0)*	0.0–0.0	*0 (0.0)*	0.0–0.0	*0 (0.0)*	0.0–0.0	*395 (10.7)*	7.7–13.6	*395 (10.7)*	7.7
*R03CC02*	*Salbutamol*	*20 (0.5)*	0.0–1.2	*12 (0.3)*	0.0–0.8	*6 (0.2)*	0.0–0.5	*17 (0.5)*	0.0–1.0	*29 (0.8)*	0.1–1.5
R03BB	Anticholinergics	198 (5.4)	2.9–7.8	171 (4.6)	2.5–6.8	165 (4.5)	2.1–6.8	146 (4)	2.0–5.9	394 (10.7)	7.4–14.0
*R03BB01*	*Ipratropium bromide*	*144 (3.9)*	1.8–6.0	*136 (3.7)*	1.8–5.5	*129 (3.5)*	1.5–5.5	*127 (3.4)*	1.6–5.3	*341 (9.2)*	6.2–12.3
*R03BB04*	*Tiotropium bromide*	*27 (0.7)*	0.0–1.7	*24 (0.6)*	0.0–1.4	*13 (0.4)*	0.0–1.1	*19 (0.5)*	0.0–1.3	*43 (1.2)*	0.1–2.2
R03AL	Adrenergics in combination with anticholinergics	120 (3.2)	1.7–4.8	94 (2.5)	1.0–4.1	108 (2.9)	1.2–4.6	141 (3.8)	2.1–5.5	305 (8.3)	5.5–10.9
*R03AL02*	*Salbutamol and ipratropium bromide*	*114 (3.1)*	1.5–4.7	*89 (2.4)*	0.9–3.9	*73 (2)*	0.7–3.2	*124 (3.4)*	1.8–4.9	*259 (7)*	4.6–9.4
R03DC (R03DC03)	Leukotriene receptor antagonists (Montelukast)	339 (9.2)	6.2–12.2	214 (5.8)	3.5–8.1	105 (2.8)	1.1–4.5	51 (1.4)	0.2–2.5	278 (7.5)	4.8–10.2
R03DX	Other systemic drugs for obstructive airway diseases	99 (2.7)	1.0–4.3	69 (1.9)	0.4–3.3	13 (0.4)	0.0–1.0	27 (0.7)	0.0–1.7	69 (1.9)	0.4–3.3
*R03DX05*	*Omalizumab*	*80 (2.2)*	0.7–3.6	*49 (1.3)*	0.1–2.5	*13 (0.4)*	0.0–1.0	*27 (0.7)*	0.0–1.7	*49 (1.3)*	0.1–2.5
*R03DX09*	*Mepolizumab*	*19 (0.5)*	0.0–1.3	*19 (0.5)*	0.0–1.3	*0 (0.0)*	0.0–0.0	*0 (0.0)*	0.0–0.0	*19 (0.5)*	0.0–1.3
**H02AB, A07EC, P01BA02,** **L03, L04AD, AX, AA, J06BA**	**Drugs to treat auto-immune diseases**	**8560 (231.7)**	**216.9–246.6**	**5104 (138.2)**	**126.8–149.6**	**3414 (92.4)**	**83.1–101.7**	**4593 (124.3)**	**113.3–135.4**	**9947 (269.3)**	**253.2–285–4**
H02AB	Systemic glucocorticoids	*6831 (184.9)*	171.7–198.2	*3673 (99.4)*	89.7–109.2	*2309 (62.5)*	54.9–70.1	*3375 (91.4)*	81.9–100.9	*8237 (223)*	208.3–237.7
L04AD, AX, AA, A07EC, P01BA02, L01XC02	Traditional immunosuppressants	1647 (44.6)	38.1–51.0	1466 (39.7)	33.6–45.8	1295 (35.1)	29.3–40.8	1338 (36.2)	30.4–42.0	1951 (52.8)	45.8–59.9
*A07EC02*	*Mesalazine*	*549 (44.6))*	11.1–18.6	*541 (14.6)*	10.9–18.4	*552 (14.9)*	11.2–18.7	*623 (16.9)*	12.9–20.9	*845 (22.9)*	18.2–27.5
*A07EC01*	*Sulfasalazine*	*197 (5.3)*	3.0–7.6	*187 (5.1)*	2.8–7.3	*146 (4)*	2.1–5.8	*128 (3.5)*	1.7–5.2	*203 (5.5)*	3.2–7.8
*P01BA02*	*Hydroxychloroquine*	*324 (8.8)*	6.1–11.5	*320 (8.7)*	5.9–11.4	*300 (8.1)*	5.4–10.8	*333 (9)*	6.1–11.9	*404 (10.9)*	7.8–14.1
*L04AX0*	*Azathioprine*	*490 (13.3)*	9.6–17.0	*384 (10.4)*	7.3–13.5	*376 (10.2)*	6.9–13.5	*345 (9.3)*	6.4–12.3	*484 (13.1)*	9.4–16.8
*L04AD02*	*Tacrolimus*	*41 (1.1)*	0.1–2.1	*38 (1)*	0.1–1.9	*47 (1.3)*	0.2–2.3	*47 (1.3)*	0.2–2.3	*47 (1.3)*	0.2–2.3
*L04AD01*	*Ciclosporin*	*6 (0.2)*	0.0–0.5	*9 (0.2)*	0.0–0.7	*0 (0.0)*	0.0–0.0	*13 (0.4)*	0.0–0.8	*21 (0.6)*	0.0–1.2
*L04AX03, L01BA01*	*Methotrexate*	*70 (1.9)*	0.5–3.3	*13 (0.4)*	0.0–1.0	*0 (0.0)*	0.0–0.0	*11 (0.3)*	0.0–0.7	*24 (0.6)*	0.0–1.4
*L04AX03*	*Methotrexate (low dose)*	*20 (0.5)*	0.0–1.3	*13 (0.4)*	0.0–1.0	*0 (0.0)*	0.0–0.0	*11 (0.3)*	0.0–0.7	*24 (0.6)*	0.0–1.4
*L01BA01*	*Methotrexate (high dose)*	*51 (1.4)*	0.2–2.5	*0 (0.0)*	0.0–0.0	*0 (0.0)*	0.0–0.0	*0 (0.0)*	0.0–0.0	*0 (0.0)*	0.0–0.0
*L04AA06*	*Myophenolic acid*	*11 (0.3)*	0.0–0.8	*11 (0.3)*	0.0–0.8	*0 (0.0)*	0.0–0.0	*0 (0.0)*	0.0–0.0	*11 (0.3)*	0.0–0.9
*L04AA29*	*Tofacitinib*	*8 (0.2)*	0.0–0.6	*20 (0.5)*	0.0–1.3	*0 (0.0)*	0.0–0.0	*0 (0.0)*	0.0–0.0	*20 (0.5)*	0.0–1.3
L04AB, AC	Biologic DMARDS	821 (22.2)	17.5–27	682 (18.5)	14.3–22.7	438 (11.9)	8.6–15.1	316 (8.6)	5.7–11.4	725 (19.6)	15.3–24.0
L04AB	TNF-alpha inhibitors	764 (20.7)	16.1–25.3	661 (17.9)	13.8–22.0	430 (11.6)	8.4–14.9	316 (8.6)	5.7–11.4	704 (19.1)	14.8–23.3
*L04AB05*	*Certolizumab*	*143 (3.9)*	2.0–5.7	*164 (4.4)*	2.4–6.4	*156 (4.2)*	2.3–6.2	*146 (4)*	2.1–5.8	*228 (6.2)*	3.8–8.5
*L04AB02*	*Infliximab*	*207 (5.6)*	3.3–7.9	*177 (4.8)*	2.8–6.8	*148 (4)*	2.2–5.8	*96 (2.6)*	1.1–4.2	*177 (4.8)*	2.7–6.8
*L04AB01*	*Etanercept*	*174 (4.7)*	2.5–7.0	*131 (3.5)*	1.8–5.3	*55 (1.5)*	0.4–2.6	*25 (0.7)*	0.0–1.5	*149 (4)*	2.1–5.9
*L04AB04*	*Adalimumab*	*179 (4.8)*	2.4–7.3	*158 (4.3)*	2.0–6.5	*91 (2.5)*	0.8–4.1	*38 (1)*	0.0–2.1	*177 (4.8)*	2.4–7.2
*L04AB06*	*Golimumab*	*66 (1.8)*	0.4–3.1	*30 (0.8)*	0.0–1.7	*10 (0.3)*	0.0–0.8	*10 (0.3)*	0.0–0.8	*30 (0.8)*	0.0–1.7
L04AC	Interleukin inhibitors	64 (1.7)	0.4–3.0	30 (0.8)	0.0–1.8	8 (0.2)	0.0–0.6	0 (0.0)	0.0–0.0	30 (0.8)	0.0–1.8
*L04AC07*	*Tocilizumab*	*45 (1.2)*	0.1–2.3	*23 (0.6)*	0.0–1.5	*0 (0.0)*	0.0–0.0	*0 (0.0)*	0.0–0.0	*23 (0.6)*	0.0–1.5
*L04AC05*	*Ustekinumab*	*8 (0.2)*	0.0–0.6	*8 (0.2)*	0.0–0.6	*8 (0.2)*	0.0–0.6	*0 (0.0)*	0.0–0.0	*8 (0.2)*	0.0–0.6
*L04AA33*	*Vedolizumab*	*17 (0.5)*	0.0–1.1	*16 (0.4)*	0.0–1.0	*6 (0.2)*	0.0–0.5	*6 (0.2)*	0.0–0.5	*16 (0.4))*	0.0–1.0
L01XC02, L04AA26	B cell therapy	*6 (0.2)*	*0.0–0.5*	*0 (0.0)*	*0.0–0.0*	*0 (0.0)*	*0.0–0.0*	*0 (0.0)*	*0.0–0.0*	*0 (0.0)*	*0.0–0.0*
*L01XC02*	*Rituximab*	*6 (0.2)*	*0.0–0.5*	*0 (0.0)*	*0.0–0.0*	*0 (0.0)*	*0.0–0.0*	*0 (0.0)*	*0.0–0.0*	*0 (0.0)*	*0.0–0.0*
*L04AA26*	*Belimumab*	*0 (0.0)*	*0.0–0.0*	*0 (0.0)*	*0.0–0.0*	*0 (0.0)*	*0.0–0.0*	*0 (0.0)*	*0.0–0.0*	*0 (0.0)*	*0.0–0.0*
J06BA02	Immunoglobulins. normal for intravascular administration	23 (0.6)	0.0–1.4	12 (0.3)	0.0–0.8	40 (1.1)	0.3–1.9	88 (2.4)	0.9–3.9	101 (2.7)	1.1–4.3
L04AA23, L03AX13, L03AB07, L04AX07, L04AA27	Multiple sclerosis specific drugs	538 (14.6)	10.9–18.3	303 (8.2)	5.7–10.8	40 (1.1)	0.1–2.0	48 (1.3)	0.4–2.3	316 (8.6)	*5.9–11.2*
*L04AA23*	*Natalizumab*	*163 (4.4)*	2.5–6.3	*111 (3)*	1.4–4.6	*9 (0.2)*	0.0–0.7	*0 (0.0)*	0.0–0.0	*111 (3)*	1.4–4.6
*L03AX13*	*Glatiramer acetate*	*169 (4.4)*	2.6–6.5	*94 (2.5)*	1.1–4.0	*31 (0.8)*	0.0–1.7	*30 (0.8)*	0.0–1.6	*100 (2.7)*	1.2–4.2
*L03AB07*	*Interferon beta-1 a*	*182 (4.9)*	2.5–7.3	*74 (2)*	0.8–3.2	*0 (0.0)*	0.0–0.0	*13 (0.4)*	0.0–0.8	*80 (2.2)*	0.9–3.4
*L04AX07*	*Dimethylfumarate*	*41 (1.1)*	0.2–2.0	*25 (0.7)*	0.0–1.3	*0 (0.0)*	0.0–0.0	*6 (0.2)*	0.0–0.5	*25 (0.7)*	0.0–1.3
*L04AA27*	*Fingolimod*	*27 (0.7)*	0.0–1.6	*14 (0.4)*	0.0–1.1	*0 (0.0)*	0.0–0.0	*19 (0.5)*	0.0–1.3	*19 (0.5)*	0.0–1.3
**J05AR + J05AE01/03/08/10 + J05AF05/07/11 + J05AG01/04/05 + J05AX08/12**	**HIV antiretrovirals**	** *477 (12.9)* **		** *483 (13.1)* **		** *610 (16.5)* **		** *671 (18.2)* **		** *760 (20.6)* **	
*J05AR03*	*Tenofovir disoproxil and emtracitabine*	*223 (6.0)*	*3.7–8.4*	*206 (5.6)*	*(3.4–7.8)*	*235 (6.4)*	*4.1–8.6*	*267 (7.)2*	*4.8–9.7*	*304 (8.2)*	*5.6–10.9*
*J05AE03*	*Ritonavir*	*112 (3.0)*	*1.5–4.5*	*99 (2.7)*	*1.4–4.0*	*179 (4.8)*	*2.9–6.8*	*192 (5.2)*	*3.2–7.2*	*205 (5.5)*	*3.5–7.6*
**D10BA, D10AD**	**Retinoids for acne treatment**	**1641 (44.4)**	** *38.1–50.8* **	** *591 (16.0)* **	** *12.3–19.7* **	** *136 (3.7)* **	** *1.9–5.5* **	** *105 (2.8)* **	** *1.1–4.6* **	** *789 (21.4)* **	** *16.9–25.8* **
*D10BA01*	*Isotretinoin (oral)*	153 (4.1)	*2.2–6.0*	*52 (1.4)*	*0.2–2.6*	*0 (0.0)*	*0.0–0.0*	*0 (0.0)*	*0.0–0.0*	*52 (1.4)*	*0.2–2.7*
*D10AD*	*Topical retinoids*	1488 (40.3)	*34.2–46.3*	*539 (14.6)*	*11.1–18.1*	*136 (3.7)*	*1.9–5.5*	*105 (2.8)*	*1.1–4.6*	*737 (20.0)*	*15.7–24.2*

Bold is to highlight the three character ATC drug groups.

### 3.2. Thyroid Therapy

The most frequently dispensed drug group was drugs to treat thyroid diseases (673.4/10,000 pregnancies). Levothyroxine was dispensed in 655.9/10,000 pregnancies (Table 2). Antithyroid preparations were dispensed in 13.3/10,000 pregnancies, most frequently propylthiouracil (7.7/10,000 in T1), followed by carbimazole (4.7/10,000 in T1, Table 3).

#### 3.2.1. Anti-Hypertensive Drugs

Exposure to anti-hypertensive drugs during pregnancy was 530.5/10,000 pregnancies, with an increase between T1 (62/10,0000) and T3 (465/10,000). Nifedipine (390.5/10,000 in T3) was the most frequently recorded anti-hypertensive, followed by beta-blockers (77.9/10,000 in T3) (Table 2). Drugs acting on the renin–angiontensin–aldosterone system (RAAS) were dispensed in 2.8/10,000 pregnancies in T2 and in 1.9/10,000 pregnancies in T3 (Table 4).

#### 3.2.2. Psychotropic Drugs

The prevalence of exposure to psychotropic drugs during pregnancy was 381.3/10,000 pregnancies. Antidepressants were dispensed in 247.7/10,000 pregnancies, most frequently selective serotonin reuptake inhibitors (SSRIs, 161.1/10,000). Exposure to any antidepressant continuously decreased from 255.6/10,000 in pre-pregnancy to 106.7/10,000 in T3.

Claims for antipsychotics were recorded during 38.6/10,000 pregnancies, with the largest decrease between pre-pregnancy (50.9/10,000) and T1 (29.8/10,000). Quetiapine (17.6/10,000) was the most frequently dispensed anti-psychotic. Lithium was claimed during 0.5/10,000 pregnancies in pre-pregnancy and in 0.5/10,000 pregnancies prior to gestational week (GW) 10 (Table 3).

Benzodiazepine derivatives were dispensed during 166.3/10,000 pregnancies, most frequently lorazepam (76.2/10,000). Claims were recorded similarly often in each pregnancy trimester. Zolpidem (40.2/10,000) was the most frequently dispensed benzodiazepine related drug.

### 3.3. Anti-Diabetics

Anti-diabetics were claimed during 389.8/10,000 pregnancies. Insulin was the most frequently claimed anti-diabetic, with increasing exposure between T1 (43.3/10,000) and T3 (331.9/10,000). Other blood glucose-lowering drugs (mainly metformin) were claimed in 60.7/10,000 pregnancies, most frequently during T1 (48.8/10,000) and decreased until T3 (16.5/10,000). The proportion of women receiving metformin to treat polycystic ovary syndrome is unknown.

#### 3.3.1. Obstructive Airway Diseases

Drugs for obstructive airway diseases were dispensed during 363.2/10,000 pregnancies, with a decrease between pre-pregnancy (198.5/10,000) and T1 (156.3/10,000, equally frequent in later trimesters). Inhaled combinations of adrenergics with glucocorticoids were the most frequently dispensed drug class (214.6/10,000) during pregnancy, followed by selective B2 adrenoreceptor agonists (161/10,000) and inhaled glucocorticoids monopreparations (54.4/10,000).

Omalizumab and mepolizumab, for which data on teratogenicity is very limited, were recorded during 1.9/10,000 pregnancies.

#### 3.3.2. Drugs to Treat Auto-Immune Diseases

##### Systemic Glucocorticoids and Traditional Immunosuppressants

Systemic glucocorticoids were dispensed during 223/10,000 pregnancies, but the proportion used to treat auto-immune diseases is unknown. Among traditional immunosuppressants, intestinal anti-inflammatories (mesalazine 22.9/10,000, sulfasalazine 5.5/10,000) were most frequently claimed, followed by aziathropine (13.1/10,000) and hydroxychloroquine (10.9/10,000). Methotrexate (oral and iv) was claimed in 1.9/10,000 pregnancies in pre-pregnancy and in 0.4/10,000 pregnancies during the high-risk period of T1. Mycophenolic acid was recorded in 0.3/10,000 pregnancies both in pre-pregnancy and T1 (none after). Cyclophosphamide was not dispensed during pre-pregnancy or T1, but it was claimed in 0.2/10,000 and 0.3/10,000 pregnancies in T2 and T3 (Table 3).

JAK kinase inhibitors (all tofacitinib) were dispensed during 0.2/10,000 and 0.5/10,000 pregnancies in pre-pregnancy and T1. There was no dispensing recorded for leflunomide (Table 5).

##### Biologic Immunosuppressants and Intravenous Immunoglobulins (IVIG)

Tumour necrosis factor-alpha (TNF-alpha) inhibitors were dispensed during 19.1/10,000 pregnancies, most commonly certolizumab (6.2/10,000). Exposure to all TNF-alpha inhibitors, except for certolizumab, decreased between T1 and T3. Interleukin inhibitors were dispensed during 0.8/10,000 pregnancies. Rituximab was only dispensed in pre-pregnancy, with 0.8 dispensations/10,000 pregnancies. Intravenous immunoglobulins were recorded during 2.7/10,000 pregnancies with an increase between T1 (0.3/10,000), T2 (1.1/10,000), and T3 (2.4/10,000).

#### 3.3.3. Drugs to Treat Multiple Sclerosis

Drugs used to treat multiple sclerosis were dispensed during 8.6/10,000 pregnancies, with a decrease between pre-pregnancy (14.6/10,000), T1 (8.2/10,000) and T2 (1.1/10,000). Natalizumab was the most frequently recorded drug during pregnancy (3/10,000) and was mainly dispensed in T1, followed by glatiramer acetate (2.7/10,000) and interferon beta (INFb) (2.2/10,000). Fingolimod was recorded during 0.7/10,000 and 0.4/10,000 pregnancies in pre-pregnancy and T1. Dimethylfumarate was recorded in 0.7/10,000 pregnancies in T1. No dispensing was recorded for mitoxantrone, teriflunomide, cladribine or alemtuzumab during pre-pregnancy or pregnancy (Table 5).

### 3.4. Human Immunodeficiency Virus (HIV) Antiretrovirals

Antiretrovirals were recorded during 15.2/10,000 pregnancies, with stable exposure across trimesters. Most frequently, the combination of tenofovir, disoproxil and emtricitabine was dispensed (8.2/10,000) followed by lopinavir with ritonavir (2.6/10,000). There were no records for stavudine or didanosine.

### 3.5. Retinoids

Retinoids were claimed during 21.4/10,000 pregnancies including 1.4 oral retinoids (4.1/10,000 in pre-pregnancy, 1.4/10,000 in T1) and 20.0 topical retinoids (40.3/10,000 in pre-pregnancy, 14.6/10,000 in T1).

**Table 3 ijerph-19-01456-t003:** Exposure to drugs with known or potential adverse reaction during trimester 1 (weighted numbers).

Drugs with Known or Potential Adverse Reaction (ATC Code)	Warnings Regarding Use during Critical Period	Risk Period	Pregnancies Exposed during Risk Period during Study Period (N)	Pregnancies Exposed during Risk Period during Study Period (%, 95%CI)
Methotrexate (L01BA01, L04AX03)	Risk of abortion. Malformations including face, skull, central nervous system, limbs, heart.	T1 (maximal risk between 8–10 GW)	13	0.4 (0.0–1.0/10,000)
Mycophenolic acid (L04AA06)	Risk of abortion. Malformations including cleft palate, ear deformities in up to 25% of exposed foetuses.	T1	11	0.3 (0.0–0.8/10,000)
Cyclophosphamide (L01AA01)	Malformations including the face, skull, eyes and central nervous system. Hematologic anomalies (anemia, leucopenia) in the newborn.	T1	0	0.0 (0.0–0.0/10,000)
Carbimazole (H03BB01)	Malformations including aplasia cutis, choanal atreasia, tracheo-esophageal fistula in up to 4% of exposed foetuses.	T1 (maximal risk between 6–10 GW)	174	4.7 (2.7–6.7/10,000)
Lithium (N05AN01)	Increased risk of heart malformation, especially the Ebstein malformation in 2.5% of exposed babies (including 1% of Ebstein disease vs. 0.005%).	T1 (until GW 10)	19.6	0.5 (0.0–0.1/10,000)
Dolutegravir (J05AR13) ^1^	Suspicion of an increased risk of neural tube defects.	T1 (Maximal risk between 0–4 GW)	16	0.4 (0.4–1.0/10,000)
Isotretinoin	Malformations including the heart, central nervous system, face, ear, eyes and thymus in up to 20% of exposed foetuses.	T1	52	1.4 (0.2–2.6/10,000)

^1^ The body of evidence on toxicity for dolutegravir is yet small.

**Table 4 ijerph-19-01456-t004:** Exposure to drugs with known or potential adverse reaction during trimester 2/3 (weighted numbers).

Drugs with Known or Potential Adverse Reaction (ATC Code)	Warnings Regarding Use during Critical Period	Risk Period	Pregnancies Exposed during Risk Period during Study Period (N)	Pregnancies Exposed during Risk Period during Study Period/10,000 95%CI)
Agents acting on the RAA system. (C09)	Renal toxicity which can induce oligo/anamnios, with potential pulmonary, skull hypoplasia and reduction of the extremities. Neonatal renal insufficiency.	T2 and T3	105, 72	2.8 (1.1–4.5/10,000) 1.9 (0.6–3.2/10,000)
Psychotropic drugs - Antidepressants; - Antipsychotics; - Benzodiazepines and related drugs (N06A, N05A, N05AN, N05BA, N05CD, N05CF)	-1/3 newborns present transient neonatal symptoms (serotoninergic toxicity or weaning symptoms); - Risk of transient extra-pyramidal symptoms in the newborn; - Prenatal exposure to benzodiazepines with long half-life in may cause impregnation symptoms (sleepiness, suction problems, hypotension, respiratory distress), whereas short half-life substances may cause weaning symptoms (agitation, hyperexcitability).	T3	779	172.2/10,000
Rituximab (L01XC02)	Neonatal B cell depletion	T2 and T3	0, 0	0.0 (0.0–0.0/10,000)
Belimumab (L04AA26)	Neonatal B cell depletion	T2 and T3	0, 0	0.0 (0.0–0.0/10,000)

**Table 5 ijerph-19-01456-t005:** Exposure to drugs with potential adverse reaction during trimester 1 (risk found in animal studies but not in human data) during risk period and associated potential risks (weighted numbers).

Drugs with Potential Adverse Reaction (ATC Code)	Warnings Regarding Use during Critical Period	Risk Period	Pregnancies Exposed during Risk Period during Study Period (N)	Pregnancies Exposed during Risk Period during Study Period (%, 95%CI)
Propylthiouracil (H03BA02)		T1	285	7.7 (5–10.5/10,000)
Mitoxantrone (L01DB07)	Contraception is recommended 4 months after the last administration.	Pre-pregnancy and T1.	0, 0	0.0 (0.0–0.0/10,000)
Fingolimod (L04AA27)	Contraception is recommended 2 months after the last administration.		27, 14	0.7 (0.0–1.6/10,000) 0.4 (0.0–1.1/10,000)
Teriflunomide (L04AA31)			0, 0	0.0 (0.0–0.0/10,000)
Leflunomide (L04AA13)			0, 0	0.0 (0.0–0.0/10,000)
Dimethylfumarate (L04AX07)	Contraception is recommended 3.5 months after the last administration.		41, 25	1.1 (0.2–2.0/10,000), 0.7 (0.0–1.3/10,000)
Cladribine (L01BB04, L04AA40)			0, 0	0.0 (0.0–0.0/10,000)
Alemtuzumab (L04AA34)	Contraception is recommended 4 months after the last administration.		0, 0	0.0 (0.0–0.0/10,000)
Tofacitinib (L04AA29) [22]			8, 20	0.2 (0.2–0.6/10,000), 0.5 (0.2–1.3/10,000)
Baricitinib (L04AA37) [23]			0, 0	0.0 (0.0–0.0/10,000)

## 4. Discussion

The aim of this claims-based study was to evaluate the utilisation of prescription drugs during pregnancy dispensed in outpatient care in Switzerland between 2014 and 2018. We focused on drugs to treat specific chronic diseases, which typically affect women of childbearing age.

### 4.1. Thyroid Therapy

Thyroid preparations were the most frequently claimed drug group during pregnancy in our cohort (673.4/10,000, 6.7%), mostly thyroid hormones (651.1/10,000). It remains to be investigated why we observed an almost two-fold higher exposure than was reported from other countries, including Norway (3.5%, national claims data) [24] and France [25] (3.1%, claims data). Untreated hypothyroidism has been associated with adverse obstetrical (spontaneous abortion, pre-eclampsia, preterm birth) and neonatal outcomes (low birth weight, impaired neuropsychological development) [26], and thus international recommendations stress the importance to pursue thyroid hormone replacement during pregnancy [26].

In our cohort, we observed decreasing exposure to the thyreostatic carbimazole between pre-pregnancy and T1 and an increase in the number of pregnancies exposed to PTU over that same period, suggesting prescribers may have advised a therapy switch in anticipation of pregnancy or when pregnancy came to medical attention. Carbimazole has been associated with specific congenital malformations in 2–4% (cutis aplasia, choanal atresia, esophageal atresia, facial dysmorphia, abdominal wall defects, nipple hypo/aplasia) [27,28]. Studies on the risk of malformations in association with propylthiouracil (PTU) have shown conflicting results [29,30,31]. The American Thyroid Association recommends PTU over carbimazole, during pre-pregnancy and T1 [32]; however, switching of drugs from carbimazole to PTU should ideally be completed before pregnancy, because poor control of thyroid function during early pregnancy is an individual risk factor of malformations [29]; however, we cannot determine how many women switched drugs earlier since the pre-pregnancy period was restricted to 3 months [32].

### 4.2. Hypertension

International guidelines recommend calcium channel blockers and beta-blockers (labetalol) as first-line antihypertensives during pregnancy [33,34], which was reflected in our cohort. Nifedipine was the most frequently claimed anti-hypertensive drug, most frequently in T3. It is often used off-label as a tocolytic drug in the management of preterm labour and is usually prescribed for 48 h in an inpatient setting; however, clinical use of nifedipine is heterogenous in Switzerland and some physicians reportedly also prescribe it for a longer period fo time in an outpatient setting. Thus, the high exposure to nifedipine in T3 in our cohort is likely partly or fully explained by its use as a tocolytic drug [35]. Methyldopa is also recommended as a first-line treatment of hypertension during pregnancy; however, we only observed dispensations in 0.6/10,000 pregnancies, possibly due to poor tolerability [36]. In utero exposure to drugs acting on the RAAS in T2 or T3 increases the risk of kidney toxicity (oligo/anhydramnios) [37,38,39] Despite a clear contraindication, we observed dispensations for RAAS-inhibitors in 2.8/10,000 pregnancies in T2 and 1.9/10,000 in T3.

### 4.3. Psychotropic Drugs

Antidepressants were dispensed in 247.7/10,000 pregnancies and decreased by more than half between pre-pregnancy and T3. This may reflect cessation of non-indicated pharmacological treatment or under-treatment due to fear of adverse drug reactions. One-third of newborns exposed to an antidepressant (SSRI, NSRI, TCA) during T3 present with transient neonatal symptoms (serotonergic toxicity/weaning symptoms), and rare complications of persistent pulmonary hypertension in the newborn have been observed [40]; however, this risk should be weighed against the risk from untreated depression during pregnancy, which has been associated with intra-uterine growth restriction (IUGR), preterm birth and low birth weight [41,42], besides maternal suicidal risk. Therefore, the UK-NICE guidelines [43] recommend that SSRIs, SNRIs, and tricyclics should all be used during pregnancy if pharmacological treatment is necessary.

Antipsychotics were dispensed during 38.6/10,000 pregnancies, most frequently quetiapine and aripiprazole, which are both considered safe during pregnancy [44,45]. Transient extra-pyramidal symptoms in the newborn are possible after in utero exposure to anti-psychotic drugs close to delivery, and newborns should be carefully monitored [44]. Claims for lithium were observed during 0.5/10,000 pregnancies in pre-pregnancy and 0.5/10,000 before GW10. Lithium should only be prescribed if all other treatments were unsuccessful, especially before GW 10, due to an increased risk of cardiac (mostly Ebstein-) malformations (2.5% vs. 0.5–1% in unexposed) [46,47].

Benzodiazepines and related drugs were dispensed during 166.3/10,000 pregnancies, most frequently lorazepam. Exposure to benzodiazepines with a long half-life (usually >20 h) in T3 may cause impregnation symptoms (sleepiness, suction problems during breastfeeding, hypotension, respiratory distress), whereas short half-life substances (usually <10 h) may cause weaning symptoms (agitation, hyperexcitability) [48]. Lorazepam has an intermediary half-life of 10–20 h. Exposure to lorazepam decreased between pre-pregnancy and trimester 2 and increased again in trimester 3. This increase may reflect heightened anxiety and possibly fear of delivery. Generally, it is important to monitor all neonates after late in utero exposure to benzodiazepines and related drugs [48].

### 4.4. Anti-Diabetics

Insulin, which is the first-line treatment of diabetes during pregnancy [49,50], was by far the most frequently claimed anti-diabetic drug during pregnancy (389.8/10,000). The strong increase in exposure of insulin between pre-pregnancy/T1 and T3 likely reflects women who were diagnosed with gestational diabetes between gestational weeks 24 and 28 [51]. Metformin was the most frequently claimed oral anti-diabetic in our cohort, followed by glibenclamide, which are the only two oral anti-diabetic drugs that may be used during pregnancy if clinically necessary [52,53]. Only 5.3/10,000 pregnancies claimed metformin continuously between pre-pregnancy and the end of pregnancy, whereas 61.1/10,000 pregnancies who have prescribed metformin before pregnancy stopped either before or during T1 or T2. While some of these women may have had diabetes and may have switched to insulin, others might have used it to treat polycystic ovary syndrome and discontinued it once they were pregnant.

### 4.5. Obstructive Airway Diseases

The proportion of pregnancies exposed to drugs for obstructive airway diseases (3.6%) was higher than the proportion of women with diagnosed asthma/chronic lung disease in the Danish study (1.73%) or in the German SNIP survey (2.7% with asthma only). Claims in our cohort likely include claims for diseases other than asthma, including inhaled glucocorticoids for allergic rhinitis. Exposure decreased between pre-pregnancy and T1 and remained stable thereafter. It has previously been reported that 1/3 of women with asthma improve during pregnancy [54]; however, we cannot determine the reason for treatment cessation, and most drugs to treat asthma are deemed safe to use during pregnancy. Untreated asthma, on the other hand, has been associated with intra-uterine growth restriction, preterm labour, and low birth weight [55], Thus, prescribers should encourage pregnant women to pursue their treatment during pregnancy whenever needed.

#### 4.5.1. Drugs to Treat Auto-Immune Diseases

##### Glucocorticoids and Nzon-Biologic Immunosuppressants

Auto-immune diseases have a strong preponderance among women of childbearing age. Both the diseases and their treatment can cause complications during pregnancy [56,57], and moreover, hormonal changes during pregnancy can lead to decreased or increased disease activity during and/or after pregnancy [58]. Claims for traditional non-biologic immunosuppressants (excluding systemic glucocorticoids) were recorded during 52.8/10,000 pregnancies, with stable exposure prevalence across trimesters. Intestinal anti-inflammatories (sulfasalazine), which are generally considered safe to use during pregnancy [59] were most frequently claimed, mirroring the high prevalence of Crohn’s disease and ulcerative colitis among auto-immune disorders. Azathioprine (13.1/10,000) and hydroxychloroquine (10.9/10,000) are also considered safe [59], and were the most frequently claimed systemic immunosuppressants in our cohort. Mycophenolate mofetil is a strong teratogen (reported malformation risk up to 20%) [60], which was claimed during 0.3/10,000 pregnancies in pre-pregnancy and 0.3/10,000 in T1. Low-dose methotrexate (reported risk of malformations 6%) [61], was claimed during pregnancies in T1, whereas high-dose intravenous methotrexate (unknown percentage of malformation risk) was only applied during pre-pregnancy (1.4/10,000). Given that none of the exposed women had recorded claims for concomitant chemotherapy, it can be assumed that methotrexate was applied off-label to abort ectopic pregnancies.

Use of systemic glucocorticoids decreased between pre-pregnancy (184.9/10,000), T1 (99.4/10,000), and T2 (62.5/10,000) and then increased again in T3 (91.4/10,000). Given that we only observe outpatient claims, this increase is unlikely to reflect use for the management of preterm labour [62]. Systemic glucocorticoids are generally considered safe for the embryo/foetus during pregnancy but should be administered at the lowest dose and for the shortest time necessary [63]. Chronic use of glucocorticoids may lead to premature rupture of the membranes and intra-uterine growth restriction. Furthermore, it increases the maternal risk of pregnancy-induced hypertension, gestational diabetes, osteoporosis, and infection [64]. Animal studies and early observational studies reported a potentially increased risk of orofacial clefts in the newborn, which was, however, not confirmed in later studies [63].

##### Biologic Immunosuppressants and Intravenous Immunoglobulins (ivig)

In total, 18.1/10,000 pregnancies were exposed to biologics in T1. TNF-alpha inhibitors were used most frequently (17.2/10,000 in T1), and exposure during pregnancy decreased for all, except for certolizumab, which remained stable across trimesters. IgG type antibodies start actively crossing the placenta around gestational week 20 via the FC fragment placental receptor [65]. Certolizumab has several indications including rheumatoid arthritis and Crohn’s disease. Clinical studies have shown low placental transfer of Certolizumab in T3, suggesting it may be safer than other TNF-inhibitors [66]. For all other TNF-alpha inhibitors, it has been proposed to administer the last dose before the beginning of T3; however, due to their long half-life, the newborn should be considered as immunodepressed until 6 months after the last administration [67,68,69]. An increased risk of malformations has not been reported in association with TNF-alpha inhibitors, and clinical situations may require continued therapy throughout T3 if benefits outweigh the expected risks. The low prescription of interleukin inhibitors (tocilizumab, ustekinumab, vedolizumab) reflects the very scarce data available for this drug group, although no teratogenic risk has been observed until now.

Rituximab (potential neonatal B cell depletion if administered after T1 [70]) was administered to 6/10,000 pregnancies during pre-pregnancy, and to none during pregnancy.

The increasing exposure to non-specific human IVIG during pregnancy may reflect treatment of exacerbations of auto-immune diseases, of specific alloimmune processes either related to the pregnancy (e.g., foetal alloimmune thrombocytopenia, severe allo-immune foetal anemia) or to other maternal diseases (e.g., organ transplant rejection) [71], or finally of infections or immunodeficiencies (congenital CMV). IVIG, similarly to the mother’s own immunoglobulins, start to increasingly cross the placenta around GW 20, but no adverse effects for the child have been reported.

#### 4.5.2. Drugs to Treat Multiple Sclerosis

We observed a strong decrease in exposure to drugs used to treat multiple sclerosis (MS) between pre-pregnancy and T3. This may reflect the fact that disease activity of MS often declines, especially during T3 [72]. Natalizumab was the most frequently claimed drug to treat MS in T1 (111/10,000) and was claimed minimally in T2 and not in T3. Natalizumab is a monoclonal antibody and should therefore not cross the placenta during T1. On the other hand, one-third of newborns exposed to natalizumab later in pregnancy have demonstrated reversible anemia, thrombopenia, and leucocytosis. [73]. Glatiramer acetate (GA) and interferon beta (INFb) are among the oldest drugs used to treat MS and are considered safe to use throughout pregnancy if needed. In our cohort, GA and INFb were the second most frequently claimed MS drugs in T1, and the most frequent ones in T2 and T3.

Mitoxantrone, a suspected teratogen [74], can be used to treat severe MS relapse [75]. No claim for mitoxantrone was reported in our cohort. Teriflunomide [76], fingolimod [77], dimethylfumarate [78], cladribine [79] and alemtuzumab [80] have previously been shown to be teratogenic in animal studies, but this risk has not been confirmed in humans yet. They should be avoided during and shortly before pregnancy, especially during T1, unless better alternatives are lacking. In our cohort, less than 5 pregnancies were exposed to either dimethylfumarate or fingolimod both during pre-pregnancy and T1. No pregnancy was exposed to teriflunomide, cladribine or alemtuzumab.

### 4.6. HIV Antiretrovirals

Overall, claims for HIV antiretrovirals during pregnancy (0.21%) were higher than the prevalence of HIV reported in Denmark (0.01%). Incidence of HIV new cases in 2017 and 2018 was 5.3/100,000 and 5.0/100,000 in Switzerland [81,82] and 3.2/100,000 and 2.7/100,000 in Denmark [83]. Recommendations regarding the management of pregnant women with HIV have recently been updated by the Panel on Treatment of Pregnant Women with HIV Infection and Prevention of Perinatal Transmission [84]. Overall, the use of antiretroviral therapy during pregnancy has not been associated with an increased risk of congenital malformation and initiation or maintenance of viral load suppression is essential to prevent materno–foetal transmission. Thus, it is strongly recommended that women pursue effective treatment throughout pregnancy except for stavudine and didanosine, which have been associated with cases of foetal lactic acidosis during pregnancy [85]. In our cohort, there were no prescriptions for stavudine or didanosine. The most frequently used antiretrovirals were the tenofovir disoproxil and emtricitabine combination (8.2/10,000) followed by ritonavir (5.5/10,000).

Despite a potentially slightly increased risk of neural tube defects if taken during T1 (WHO warning in 2018), dolutegravir has recently been recommended as first-choice therapy during pregnancy due to its advantages including once-daily dosing, its good tolerance, its rapid and durable viral load suppression [84]. However, claims for dolutegravir in our cohort were low.

### 4.7. Retinoids

Claims for systemic oral retinoids were recorded during 1.4/10,000 pregnancies in T1. Oral retinoids are a known teratogen that can cause numerous malformations in up to 20% of foetuses exposed. According to Swissmedic [86], they are submitted to strict prescription rules which include the initiation of an efficient contraception method one month before starting the treatment as well as a negative pregnancy test. Contraception should be maintained during the whole duration of the treatment and pursued until one month after the end of treatment for isotretinoin (3 years for acitretine, for which we did not record any prescription). The prescription should be renewed every 30 days.

### 4.8. Clinical and Policy Implications

Our study shows, for the first time, how chronic diseases during pregnancy are treated in Switzerland on a national level. Knowledge of what is done in clinical practice is of great importance with regard diverse aspects of public health, such as allocation of research funding to areas of greatest medical need, introduction of safety measures on a societal level, development of reimbursement policies and many more. Furthermore, our study shows that up to 22% of pregnancies in Switzerland were exposed to drugs for chronic diseases (not including epilepsy, which was studied separately). This suggests that up to one out of five pregnancies should be considered at "high risk" for maternal complications and should therefore be followed by a materno–foetal medicine specialist, or more ideally by a multidisciplinary team.

### 4.9. Strengths and Limitations

This study, along with our previous study on the utilisation of drugs to treat acute symptoms and conditions of pregnancy, is the first to provide a complete picture of drug utilisation during pregnancy on a population-based level in Switzerland. This study provides a good reflection of the prevalence of chronic diseases in this population, given that drugs to treat chronic diseases are not typically purchased over the counter. Our database includes longitudinal data from 15% of all pregnancies in Switzerland. Mean maternal age and the proportion of caesarean sections in our weighted cohort were consistent with statistics reported by the Swiss Federal Statistical Office for the overall population in Switzerland for this time period [21]. Thus, it can be assumed that our weighted study population is representative of the overall population of pregnant women in Switzerland.

However, certain limitations should be acknowledged. Despite the large number of individuals insured with Helsana, the largest insurance company in Switzerland, our results have been weighted based on only 15% of the Swiss population. Socio-economic aspects were not available to consider in the extrapolation, and we can therefore not rule out that pregnant women insured with Helsana may differ from pregnant women insured with other insurers in terms of specific socio-economic factors. However, maternal age at delivery is a known proxy for socio-economic status [87] and was consistent with numbers by the Swiss Federal Statistical Office. Thus, substantial channelling seems unlikely. Second, some weighted numbers were extrapolated from small numbers of exposed women and need to be interpreted with caution. Further studies should aim to replicate those results using different data. Third, the prevalence of exposure to some teratogenic drugs may have been underestimated due to the absence of data on pregnancies resulting in induced abortion or miscarriage. These pregnancies were not included in our cohort since they cannot be reliably captured in Helsana claims data. Fourth, this study only evaluated outpatient drug use, since inpatient drug use is not separately reimbursed but is part of the bundled DRG-based reimbursement system for inpatient stays. A survey among Swiss obstetric clinics previously reported which drugs were routinely used to treat various pregnancy and post-partum indications in an inpatient setting [88]. Fifth, the date of the LMP, as well as pregnancy trimesters, had to be estimated using DRG codes, since gestational age at delivery is not recorded. This estimation was carried out using an algorithm validated in US claims data [15]. This algorithm has not been validated in Switzerland due to the absence of gestational age at delivery in the recorded data. This would require linkage of different external data sources. Contrary to other countries, this has not yet been possible in Switzerland due to legal and political restrictions over privacy concerns in Switzerland. Therefore, some misclassification of prescription timing by trimester is possible. Sixth, only drug dispensations are recorded, and we cannot determine whether and when the drug was actually taken by the patient. Seventh, the underlying indications for the use of drugs could not be assessed since medical diagnoses are not captured in the TARMED system. Therefore, we could not assess whether the dispensation of potentially teratogenic or fetotoxic drugs was clinically indicated.

## 5. Conclusions

Overall, patterns of claims for drugs to treat chronic diseases during pregnancy in Switzerland were in line with treatment recommendations; however, relatively rare events of in utero exposure to teratogenic drugs may have had severe implications for those involved. Healthcare claims databases appear to be a valuable tool to study drug utilisation during pregnancy in Switzerland. However, due to legal and political restrictions, they have not been used to their full potential.

## Figures and Tables

**Figure 1 ijerph-19-01456-f001:**
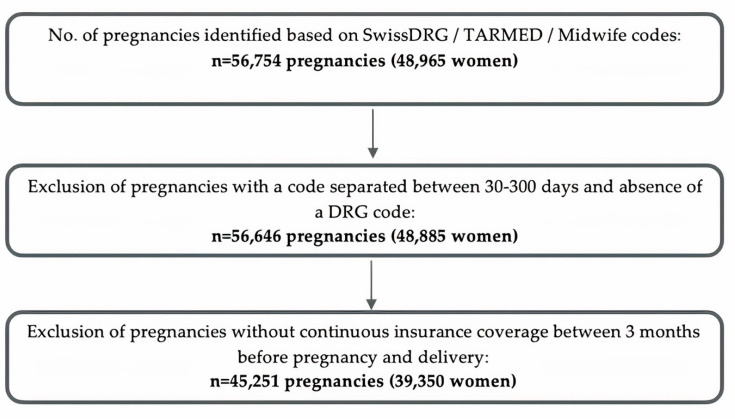
Flowchart of unweighted study population.

**Figure 2 ijerph-19-01456-f002:**
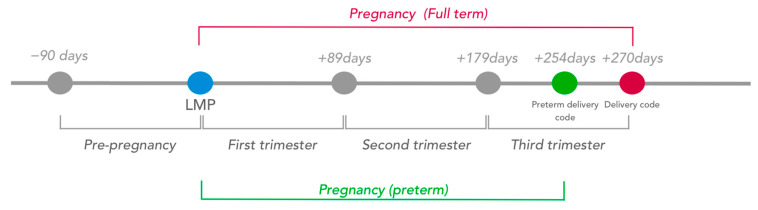
Estimation of the last menstrual period and pregnancy periods.

**Figure 3 ijerph-19-01456-f003:**
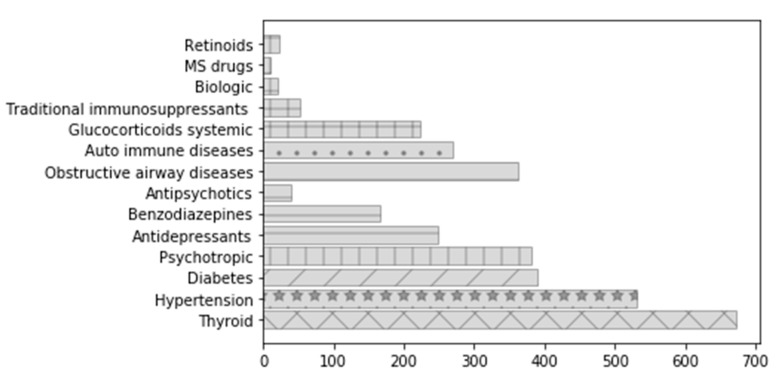
Prevalence of pregnancies per drug group during T1–T3 (Each drug group is shown with a different pattern. The group of psychotropic comprises antidepressants, benzodiazepines and anti-psychotic and is thus represented with the same pattern. The auto-immune disease group comprises glucocorticoids, traditional immunosuppressants, biologic and MS drugs).

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
