# Peer review of "Use of Prescribed Drugs to Treat Chronic Diseases during Pregnancy in Outpatient Care in Switzerland between 2014 and 2018: Descriptive Analysis of Swiss Health Care Claims Data"

_ijerph, 2022, doi:10.3390/ijerph19031456_

Round 1

Reviewer 1 Report

Interesting article and important information.

In the supplementary tables, where unweighted numbers are presented, several drugs are marked with an asterix*. This indicates that absolute numbers are below 10. If such low numbers are now extrapolated and weighted, the so-called weighted numbers have to be interpreted with caution. Please discuss this in 4.7. “Strength and Limitations” and indicate to which drugs it applies.

Points to consider:

  • Table 2 is placed before Table 1. Furthermore, I think that Table 2 should better be placed in the result section.
  • Table 3: Retinoids belong to the major teratogens. Acne is a chronic disease of younger people and isotretinoin a teratogenic drug to treat acne. Perhaps there are no cases, because all women with exposure during the risk period decided for induced abortion. However, it is worth looking for isotretinoin.
  • (Table 3: The teratogen hydroxycarbamide is mentioned on the site of Le Crat, but is missing here.)
  • Table 4 is for potential teratogens and Table 3, too. This is a bit confusing. If the authors add dolutegravir to Table 4, Table 3 would only consist of known teratogens.
  • The selection of drugs/drug groups of Table 4 is not fully understood. RAAS inhibitors okay, psychotropic drugs also okay. From the heterogeneous group of immunosuppressants two are selected. Why these two? I suppose that for example belimumab can also lead to neonatal B cell depletion.
  • Table 5: Was there no warning issued by Swissmedic regarding bosentan?
  • Typing error in the supplementary tables. I have understood that unweighted numbers are presented. Please correct.
  • P17, “4.2. Hypertension”. Tocolytic therapy is recommended for inpatients for 48 hours. Clinical experience is, however, that nifedipine therapy is often continued for a longer period and thus in outpatients. In the cited reference [34] (https://www.sggg.ch/fileadmin/user_upload/41_Tocolyse_2013.pdf) duration of nifedipine therapy as tocolytic drug is not restricted to 48 hours – in contrast to other tocolytic drugs. The only plausible explanation for the high use of nifedipine during the 3rd trimester is its use as tocolytic drug. Furthermore, nifedipine can be used for hypertension in pregnancy, but is not the first-line drug (https://www.awmf.org/uploads/tx_szleitlinien/015-018l_S2k_Diagnostik_Therapie_hypertensiver_Schwangerschaftserkrankungen_2019-07.pdf).
  • Line 319: RAAS inhibitors during T2/T3 do not increase the malformation. Malformations are induced in T1.
  • Line 326: I suggest to replace “teratogenicity” by adverse drug reactions.
  • Line 399: Mycophenolate mofetil is a major teratogen with a malformation risk of probably above 20%. However, this is not true for methotrexate. Methotrexate in low-dosages has a malformation risk of about 6%. As only outpatient drug use is considered in this study, methotrexate was probably primarily used in weekly low-dose. To my best knowledge, studies quantifying the risk of higher dosages do not exist.
  • Line 413, please add a reference
  • Line 421. The exact text of the SMPC of Cimzia® is “… However, the available clinical experience is too limited to, with a reasonable certainty, conclude that there is no increased risk associated with Cimzia® administration during pregnancy. …. Cimzia® should only be used during pregnancy if clinically needed. …” (https://www.ema.europa.eu/en/documents/product-information/cimzia-epar-product-information_en.pdf) It is true that placental transfer in the second half of pregnancy is much lower (or absent) than with other TNF-inhibitors. The wording Cimzia® ”is indicated for use during pregnancy”, however, reflects the manufacturer’s point of view.
  • Line 515: please add induced: “… resulting in induced abortion or miscarriage”

Reviewer 2 Report

The manuscript under review explored the use of prescribed drugs to treat chronic diseases during pregnancy in outpatient care in Switzerland between 2014 and 2018 using Swiss health care claims data. The study focuses on evaluate utilization of drugs to treat chronic diseases during pregnancy in Switzerland. Even new information is identified, however, the findings is weakened by several issues including incomplete description of background, method and results, inconsistency of gap identified, results and discussion, and issues in the logical presentation of the study aims. Details relating to these and other issues are presented below.

  1. Authors should state the reasons for their study in relation to chronic disease and related misuse and adverse neonatal outcomes (safety?) including and reviewing previous literature or observation. Additionally, authors should revise the last sentence…aims of this study (scope down the research…too much information in a paper).
  2. Authors should provide the challenging issues on the use of prescribed drugs ( or physicians’ prescribing drugs without considering the neonatal outcomes?) to treat chronic diseases during pregnancy in outpatient care in the context of Switzerland health system / health market/ academic research / methodological issues.
  3. line 132: Figure 3: A description of the figure should be presented in the text.
  4. The methods of the analysis for the content presented in the table 2 should be presented in the text.
  5. It may better to selectively present only important information in the tables.
  6. It may better to describe the methods/ meanings related to unweighted numbers and weighted numbers in the texts.
  7. line 348: “succion problems” refer to what?
  8. Suggest to revise the discussion part: describe more logically.
  9. line 348: “succion problems” refer to what?
  10. Suggest to revise the conclusion part: Authors should consider the valuable part of this draft; highlight any gaps related to the topic of this study.

Reviewer 3 Report

1. This study is descriptive in nature. Therefore, the authors may consider changing the subtitle to: "Descriptive Analysis of Swiss health care claims data."
2. Authors need to define acronyms in the abstract. 
3. The abstract conclusion does not demonstrate the achievement of the objectives. The authors aimed to analyze drugs prescribed for chronic diseases during pregnancy but not quantify chronic diseases' prevalence.
4. The manuscript should be revised to conform with the STROBE reporting guidelines for the study design. 
5. Authors should include a discussion section on the clinical or policy Implications of their findings. 

Round 2

Reviewer 3 Report

It should be "Clinical and policy implications" NOT " Clinical and political implications " Please replace the word 'political' with 'policy'

Author Response

The word "political" has been replaced with the word "policy".